# Scale-Aware Edge-Preserving Full Waveform Inversion with Diffusion Filter for Crosshole Sensor Arrays

**DOI:** 10.3390/s24092881

**Published:** 2024-04-30

**Authors:** Jixin Yang, Xiao He, Hao Chen, Jiacheng Li, Wenwen Wang

**Affiliations:** 1State Key Laboratory of Acoustics, Institute of Acoustics, Chinese Academy of Sciences, Beijing 100190, China; yangjixin@mail.ioa.ac.cn (J.Y.); hex@mail.ioa.ac.cn (X.H.); lijiacheng.chn@foxmail.com (J.L.); 2University of Chinese Academy of Sciences, Beijing 100049, China; 3China United Coalbed Methane Corporation, Ltd., Beijing 100011, China; wangww7@cnooc.com.cn

**Keywords:** full waveform inversion, scale-aware edge-preserving, nonlinear anisotropic hybrid diffusion filer, crosshole survey, coherence enhancing diffusion, edge enhancing diffusion

## Abstract

Full waveform inversion (FWI) is recognized as a leading data-fitting methodology, leveraging the detailed information contained in physical waveform data to construct accurate, high-resolution velocity models essential for crosshole surveys. Despite its effectiveness, FWI is often challenged by its sensitivity to data quality and inherent nonlinearity, which can lead to instability and the inadvertent incorporation of noise and extraneous data into inversion models. To address these challenges, we introduce the scale-aware edge-preserving FWI (SAEP-FWI) technique, which integrates a cutting-edge nonlinear anisotropic hybrid diffusion (NAHD) filter within the gradient computation process. This innovative filter effectively reduces noise while simultaneously enhancing critical small-scale structures and edges, significantly improving the fidelity and convergence of the FWI inversion results. The application of SAEP-FWI across a variety of experimental and authentic crosshole datasets clearly demonstrates its effectiveness in suppressing noise and preserving key scale-aware and edge-delineating features, ultimately leading to clear inversion outcomes. Comparative analyses with other FWI methods highlight the performance of our technique, showcasing its ability to produce images of notably higher quality. This improvement offers a robust solution that enhances the accuracy of subsurface imaging.

## 1. Introduction

Full waveform inversion (FWI) tomography emerges as an indispensable tool for crosshole imaging with unparalleled high resolution, providing intricate insights into subsurface structures. In the realm of CO_2_ capture, utilization, and sequestration (CCUS) (CCUS) [1,2,3], the role of crosshole surveys is gaining heightened recognition due to their indispensable contribution to the fidelity of subsurface monitoring. Foremost among these methods is the crosshole radar, renowned for its effectiveness in characterizing geological layers. The defining feature of crosshole acoustical datasets is their comprehensive broadband characteristics, replete with rich high-frequency data that afford significantly superior imaging resolution over traditional surface seismic surveys [4,5,6]. This broadband spectrum is quintessential for the detailed visualization of stratigraphic features and the monitoring of CO_2_ injection and migration processes, offering a substantial leap in the predictive modeling of geological storage capabilities and the assessment of environmental safety.

Nevertheless, FWI tomography is intrinsically a highly nonlinear inverse problem, where the outcomes of the inversion are often beset by instability [7]. This instability becomes particularly acute in the context of crosshole seismic datasets. Such datasets are acutely sensitive to various noise sources, a vulnerability stemming from the complexities of the observation apparatus and the broad frequency bandwidth employed. These factors conspire to produce waveforms that are intricate and laden with noise, within which valuable signals—chiefly those deriving from reflected and transmitted waves—are deeply ensconced [8,9]. The pervasiveness of multifarious noise sources exacerbates the nonlinearity of the FWI process, imposing formidable obstacles to attaining a stable and convergent solution. In addition, there is often a disparity between the theoretical forward models—deployed to emulate the passage of seismic waves through the Earth’s subsurface—and the actual propagation of these waves through multifaceted geological structures, a discrepancy that is frequently inescapable [10]. This gap can be attributed to a range of factors: the imprecise knowledge of the initial subsurface model, oversimplifications inherent in the physical modeling assumptions, and the intrinsic constraints of current seismic data acquisition technologies. Such inconsistencies underscore the necessity for robust inversion methodologies capable of accommodating these imperfections and delivering reliable subsurface characterization despite the inherent challenges.

To circumvent these impediments and enhance the robustness of FWI, the adoption of methodical approaches, such as regularization and model update filtering, becomes critical [11]. A plethora of strategies have been methodically devised to this end. Prominent among these is the integration of regularization techniques within the FWI algorithm, a method proving to be particularly efficacious. This approach involves the judicious incorporation of prior geological information into the inversion process, thus tethering the solution to a geologic framework [8,9,12]. Quadratic regularization methods, such as Tikhonov regularization, are not only computationally expedient, but also adept at mitigating the oscillatory artifacts frequently encountered in datasets beset by noise. In contrast, non-quadratic regularization techniques, notably total variation (TV) regularization, are esteemed for fostering parameter sparsity [11,12]. This characteristic significantly refines the demarcation of model boundaries, thus augmenting the geological verisimilitude of the resultant inversion models. The strategic application of these regularization methodologies allows FWI to deftly navigate the trade-off between computational pragmatism and the veracity of the reconstructed subsurface models. Consequently, this enhances the prospects for more nuanced and dependable geological interpretations, supporting the advancement of subsurface exploration and monitoring in geophysical research.

In the realm of FWI, the implementation of filter-based methodologies, exemplified by the utilization of Gaussian filters, is integral in the process of model update smoothing throughout successive inversion iterations, which substantially contributes to noise reduction. Yet, these filters are not without limitations. Their inherent smoothing action often inadvertently obfuscates essential structural details, thereby risking the precision of the subsurface model [13,14,15]. To mitigate such drawbacks, the field has witnessed the emergence of directional Laplacian filtering. This sophisticated technique provides selective smoothing that adeptly upholds the fidelity of structural delineations. In response to this challenge, directional Laplacian filtering has emerged as a refined solution, offering nuanced smoothing that adeptly preserves the integrity of structural features. Further advancements in this domain have seen the integration of partial differential equations (PDE)-based smoothing techniques, employing Bessel filters for their unique ability to maintain edge sharpness while smoothing [16,17]. Moreover, the application of TV regularization marks a significant advancement in maintaining edge definition within FWI. This approach has been particularly beneficial in enhancing the sharpness of contrast in complex geological features, such as salt body delineations [18,19,20,21,22,23,24,25]. Despite the strides made with these methods, they must continuously reconcile the delicate equilibrium between noise suppression and the preservation of unambiguous structural and edge clarity. This balance is critical to the iterative refinement and sophistication of FWI techniques, reflecting the dynamic progression and relentless pursuit of precision within geophysical imaging methodologies.

In this research, we unveil a sophisticated SAEP-FWI methodology, ingeniously tailored to address the intricacies inherent in crosshole tomography. At the forefront of our methodology is the integration of the NAHD filter, a cutting-edge innovation distinguished by its dual edge-enhancing and coherence-enhancing properties. The prowess of the NAHD filter lies in its adeptness at attenuating noise while meticulously conserving the detailed structural nuances present in the FWI gradient—thereby striking an optimal equilibrium between clarity and precision. Our proposed methodology augments the conventional FWI framework with Tikhonov regularization, forming a sophisticated hierarchical inversion structure that is finely tuned to mitigate the oscillatory artifacts often prevalent in crosshole dataset inversions. This paper offers an in-depth exploration of the NAHD filter’s integration into the SAEP-FWI process, presenting a comprehensive evaluation of its influence on the accuracy and stability of inversion outcomes. The robustness of our algorithm is corroborated through comprehensive numerical simulations, coupled with its successful application to empirical crosshole datasets, illustrating its capacity to markedly elevate the fidelity of subsurface imaging. These findings suggest a substantial advancement in the domain of geophysical exploration, signaling a new echelon of precision in the visualization and characterization of subsurface formations.

## 2. Methods

### 2.1. Nonlinear Anisotropic Hybrid Diffusion Filter

The diffusion filter is analogous to the physical diffusion process based on the general anisotropic diffusion equation [26]:
(1)$$\frac{\partial U}{\partial t}=\nabla \left(D\nabla U\right),$$
where ∇ is the divergence operation and D is the diffusion tensor, which can steer the diffusion. U represents the input data. We can adjust the diffusion process to coherence enhancing diffusion (CED) filtering by changing the eigenvalues of the diffusion tensor D with a structure tensor to steer the filtering for directional, anisotropic smoothing, and the structure tensor can be defined as [26]:
(2)$$J_{\rho }\left(\nabla U_{\sigma }\right)=K_{\rho }\times \left(\nabla U_{\sigma }\nabla U_{\sigma }^{T}\right),$$
where Kρ is the Gaussian kernel with standard deviation ρ, which can determine the average of the orientation information. $\nabla U_{\sigma }$ represents the gradient of the input data at scale σ, at which the derivatives for the gradient in input U are determined. The eigenvalues of Jρ, $\mu_{1}$, and $\mu_{2}$, can be used to replace the eigenvalues of diffusion tensor D. With the smoothing kernel Kρ, $\mu_{1}$ and $\mu_{2}$ describe the local contrast structure for input data. Then the eigenvalues of the CED can be written as [26]:
(3)$$\begin{array}{l} \lambda_{c1}=\alpha \\ \lambda_{c2}=\left\{\begin{array}{cc} 1, & \mu_{1}=\mu_{2}\\ \;\alpha +\left(1-\alpha \right)\cdot e^{\frac{-\ln\left(2\right)\cdot \lambda_{c}^{2}}{\kappa }}, & else \end{array}\right. \end{array},$$
where $\kappa ={\left(\mu_{1}/\left(\alpha +\mu_{2}\right)\right)}^{4}$, α=0.001 [14], and $\lambda_{c}$ is the CED contrast parameter, which can enhance different levels of flow-like structure.

In the edge enhancing diffusion (EED) filtering, the matrix D is constructed by the eigenvectors ($\vec{v_{1}}$) and ($\vec{v_{2}}$) according to $\vec{v_{1}}||\nabla U_{\sigma }$ and $\vec{v_{2}}⊥\nabla U_{\sigma }$. The corresponding eigenvalues $\lambda_{e1}$, $\lambda_{e2}$ can be written as [26]:
(4)$$\begin{array}{l} \lambda_{e1}=\left\{\begin{array}{cc} 1, & \left({\left|\nabla U_{\sigma }\right|}^{2}=0\right)\\ 1-e^{\frac{-C}{{\left({\left|\nabla U_{\sigma }\right|}^{2}/\lambda_{e}^{2}\right)}^{4}}}, & else \end{array}\right.\\ \lambda_{e2}=1 \end{array},$$
where $\lambda_{e}$ is the contrast parameter. If the squared gradient ${\left|\nabla U_{\sigma }\right|}^{2}$ much smaller than $\lambda_{e}$, isotropic diffusion will be performed. Diffusion will decrease if the squared gradient increases compared to $\lambda_{e}$, indicating a plate-like structure. The threshold parameter *C* = 3.31488 as defined in [26].

The heterogeneity of crosshole structures, encompassing a wide array of geometries, scales, and contrasts, necessitates a sophisticated approach to filtering. Employing EED in isolation can proficiently attenuate noise and underscore plate-like formations, characteristic of thin-sheet layers frequently encountered in crosshole environments; however, this method may inadvertently neglect the delineation of smaller, complex features. In contrast, the application of CED in isolation has the potential to highlight flow-like patterns and fine details, yet it also bears the risk of inadvertently reinforcing noise structures that mimic these coherent patterns. To surmount these limitations, the nonlinear anisotropic hybrid diffusion (NAHD) filter is introduced, ingeniously designed to capitalize on the strengths of both EED and CED. This hybrid filtering technique features a dynamic switching mechanism, enabling it to fluidly transition between EED and CED, thereby ensuring optimal structural enhancement and noise suppression. This adaptive filtering capability allows for a more faithful representation of the subsurface structures, providing a versatile tool for improved interpretation and analysis of geophysical data. The eigenvalues of the hybrid diffusion tensor $\lambda_{hi}$ are set as the linear combination of the eigenvalues of the EED $\lambda_{ei}$ and CED $\lambda_{ci}$:
(5)$$\lambda_{hi}=\left(1-\varepsilon \right)\cdot \lambda_{ci}+\varepsilon \cdot \lambda_{ei},$$
where ε is the EED fraction. CED will be performed when ε→0 or EED will be carried out when ε→1. Moreover, it can be written as:
(6)$$\varepsilon =e^{\frac{\mu_{1}\left(\lambda_{h}^{2}\left(\xi -\left|\xi \right|\right)-2\mu_{2}\right)}{2\lambda_{h}^{4}}},$$
(7)$$\xi =\frac{\mu_{1}}{\alpha +\mu_{2}},$$
where ξ is a parameter that can distinguish between plate-like and flow-like structures. The hybrid contrast parameter $\lambda_{h}$ controls the hybrid filtering process with EED and CED.

In this research, we set the scale σ=0.5, ρ=3.0. The hybrid contrast parameter $\lambda_{h}$ can be used for switching between the EED and CED. When adjusting parameters for FWI, the most important parameters to tune are $\lambda_{h}$, $\lambda_{e}$, and $\lambda_{c}$, which should be small enough to enhance useful flow-like structure and large enough to filter noisy speckles.

### 2.2. Scale-Aware Edge-Preserving FWI

The objective function with the Tikhonov regularization can be defined as follows [8]:
(8)$$J_{Tik}\left(m\right)=\frac{1}{2}\sum_{s,r,\omega }{\left\|d_{s,r}^{cal}\left(m,\omega \right)-d_{s,r}^{obs}\left(\omega \right)\right\|}_{2}^{2}+\eta {\left\|Lm\right\|}_{2}^{2},$$
where m is the velocity model, ω is the angular frequency, and $d^{cal}$ and $d^{obs}$ are the calculated and observed crosshole waveforms, respectively. The first term is the data misfit, and the second is the Tikhonov regularization term. η is the regularization parameter to balance the contribution from two terms. L is the first-order differencing operator along each axis dimension of the model. With the adjoint state method, the corresponding gradient can be written as follows:
(9)∇JTikm=diagHa+I−1GmRepT∂A∂mTA−1Δd*+ηLTLm,
where $\mathrm{diag}H_{a}$ represents the diagonal element of approximate Hessian $H_{a}$ and $G_{m}$ is a Gaussian filtering operator. $A=\left(\omega^{2}/v^{2}\right)-\nabla^{2}$ is a complex-valued Helmholtz matrix with respect to the frequency and model parameters. Calculated wavefield p can be obtained with the linear matrix system:
(10)Ap = s,
where s is the source vector.

Instead of applying conventional Gaussian filtering to remove the noise from FWI gradients, we can formulate the FWI gradients with the NAHD filtering $N_{m}$ and Tikhonov regularization, as follows:
(11)∇JTikm=diagHa+I−1NmRepT∂A∂mTA−1Δd*+ηNmLTLm,

Building upon the dualistic capabilities of the NAHD, a tiered approach can be conceived where the NAHD filter dynamically oscillates from CED to EED. This hierarchical strategy commences with the application of CED to expunge noise from the FWI gradient, whilst remaining cognizant of the characteristic flow-like structures, such as the crosshole thin-sheet layers. Subsequently, the NAHD filter judiciously transitions towards EED, prioritizing the preservation of sharp edges within the FWI gradient. This deliberate and sequential application of CED and EED within the NAHD framework facilitates a scale-aware edge-preserving filtering capability, which is pivotal for the enhanced fidelity of SAEP-FWI. The proposed method thus offers a robust solution for the high-precision imaging requirements of FWI, ensuring that both the subtle nuances of geological formations and the defining boundaries of subsurface features are captured with clarity and precision.

## 3. Results

### 3.1. Synthetic Experiments

To rigorously evaluate the performance of the proposed SAEP-FWI approach, we employed a multi-layer toy crosshole model as depicted in Figure 1a. The computational domain for this model is defined by a grid measuring 578 by 216, with a uniform grid spacing of 1 m in both horizontal and vertical dimensions. As illustrated in Figure 1a, a string of 192 sources was strategically placed within the source borehole, spaced 3 m apart, indicated by black pentagrams. Correspondingly, sensors were similarly spaced at 3 m in the sensor borehole, denoted by red triangles. To facilitate the initial conditions for both the FWI and SAEP-FWI processes, we applied a smoothing filter to the original crosshole model, resulting in the smoothed initial model depicted in Figure 1b. Additionally, to assess the robustness of our method under practical scenarios, we introduced random noise into the synthetic dataset, as illustrated in Figure 2. This inclusion enables a comparative analysis of the FWI results obtained using different gradient processing strategies, thereby providing a comprehensive understanding of the effectiveness of the method in realistic noisy environments. This structured approach ensures that our evaluation is both thorough and indicative of the potential of SAEP-FWI scheme under varied geological and noise conditions.

We employed ten groups of frequency components, ranging between 200 and 400 Hz, each consisting of five adjacent frequencies incrementally spanning from the lower to the higher end of this spectrum. These groups underwent a sequential inversion process where each group was subjected to a fixed number of 30 iterations, without regard for convergence. This procedure was designed to ensure consistency and computational manageability across all frequency bands. Upon completion of iterations for one group, the resulting velocity model served as the initial model for the inversion of the subsequent higher frequency group. This methodical approach facilitates the progressive refinement of the velocity model with the integration of higher-frequency data. Our analysis began with the application of NAHD filtering on the FWI gradient of the initial model depicted in Figure 1b. We selected the first group of frequencies to compare its gradients. The original gradient, displayed in Figure 3a, exhibited spatial irregularities due to the nonlinearity and instability inherent in FWI. Subsequently, conventional Gaussian filtering was applied to the gradient, as shown in Figure 3b. Although this approach smoothed the noise-like irregularities, it unfortunately failed to preserve, and even distorted, the small-scale structural details and edges. In contrast, the application of NAHD with its CED and EED filtering schemes for FWI gradients, as presented in Figure 3c,d, offers a compelling comparison. The switch to CED nearly eliminated all irregularities while preserving structural details. The main structure and small-scale layering information were enhanced through CED without obscuring useful information, a benefit attributed to the design of CED to enhance flow-like textures, which is particularly advantageous for investigating multi-layer structures in subsurface crosshole surveys. Adjusting the parameters to implement EED facilitated the smoothing of noise while accentuating edges, plate-like structures, and layering in crosshole surveys. As illustrated in Figure 3d, EED effectively removed irregularities, and edges were significantly highlighted, although the continuity of small-scale structures was somewhat diminished. Therefore, for the SAEP-FWI, we adopted a hierarchical strategy to maximize the effectiveness of the NAHD filter. In the initial stages of inversion, where lower frequency groups are primarily processed, the NAHD filter was specifically configured to operate as a CED mechanism. This configuration aimed to effectively reduce noise while simultaneously preserving small-scale structures. As the inversion progressed to higher frequency groups, this approach gradually increased the accuracy of methods; however, it also enhanced the nonlinearity of the inversion process due to the greater sensitivity to detailed structural variations. Subsequently, we adjusted the parameters by a factor, prompting a gradual shift to the EED filtering scheme to emphasize edges. This adjustment was necessary because, as the FWI iterations converged, the inversion’s nonlinearity decreased, making EED more suitable for preserving edges without compromising the continuity of small-scale structures. Subsequently, we increased the $\lambda_{e}$ parameters by a factor, prompting a gradual shift to the EED filtering scheme to emphasize edges. This is because as the FWI iterations converge, the inversion’s nonlinearity decreases, making EED more suitable for preserving edges without compromising the continuity of small-scale structures.

We evaluated four FWI strategies, the outcomes of which are presented in Figure 4. The result of conventional FWI, depicted in Figure 4a, exhibits noise-like irregularities in crosshole tomography, and the edges of small-scale structures appear distorted, misaligning with the true crosshole model. Subsequently, we examined the FWI result with Gaussian filtering. Although this method removed some noise, as shown in Figure 4b, the average filtering technique unfortunately blurred the edges, reducing structural clarity. We then implemented NAHD filtering using the CED scheme exclusively. CED is known for its ability to enhance flow-like structures and as indicated in Figure 4c, it almost entirely removed noise while preserving the integrity of crosshole structures. Nonetheless, some residual noise remained, and the edges of structures could benefit from further enhancement. To address this, we configured NAHD to transition from CED to EED as the frequency groups shifted from low to high. This strategic adjustment is illustrated in Figure 4d, where the SAEP-FWI shows significant improvements in preserving edges and eliminating irregularities, leading to a more accurate representation of the subsurface. It is critical to note the potential drawbacks of initially employing EED in the inversion process. While EED is highly effective in preserving plate-like structures and removing noise in homogeneous areas, it may impede the convergence of the inversion due to its intense focus on edge preservation, which can lead to the overemphasis of certain features at the expense of effectively capturing smaller-scale structures. Thus, the phased integration of CED and EED within NAHD is crucial for optimizing the inversion results across different frequency bands, ensuring a balanced approach to noise reduction and structural fidelity.

### 3.2. Field Dataset Application

The crosshole field dataset was acquired from two parallel boreholes spaced approximately 196 m apart. Sensors were installed with a 3.0 m spacing in one borehole, and an acoustic source was activated at 3.0 m intervals in the other borehole. Observations spanned a depth range from 0 m to 400 m. The computational domain was set at 400 m by 196 m with a grid interval of 1 m. As depicted in Figure 5, this crosshole dataset is band-limited, exhibiting a frequency band gap between 0 and 100 Hz. The effective frequency band, suitable for full waveform inversion (FWI), ranges from 100 to 400 Hz. Due to the absence of low-frequency information, the FWI process heavily relies on an accurately constructed initial model. For this crosshole FWI application, the initial model was derived from the curved ray travel time tomography results. Figure 6 illustrates the initial tomographic model used. To address the convergence issue often encountered in the real dataset, the source parameters were meticulously estimated from the real dataset in the frequency domain, enhancing the accuracy and reliability of the inversion process.

In this study, the FWI process for the real dataset utilizes frequency information spanning from 100 to 400 Hz, divided into twelve frequency groups, each containing five neighboring frequencies. We conducted a comparative analysis of four FWI strategies applied to this field dataset, as illustrated in Figure 7. Figure 7a displays the results from the conventional FWI approach, which does not incorporate any filtering or preconditioning of the FWI gradient. This conventional result captures some structural information from the crosshole but is also marred by noise-like irregularities attributed to the ambient noise in the field dataset and the inherent nonlinearity of the FWI process. Consequently, this speckle noise and these irregularities must be mitigated to enhance the quality of the crosshole FWI image. A commonly adopted method involves the introduction of a Gaussian filter in the FWI gradient formulation. This method applies a Gaussian filter to precondition the gradient prior to the application of the Hessian operator, as shown in Figure 7b. While the application of Gaussian smoothing clarifies the image by reducing noise, it unfortunately tends to smooth out small-scale structures, occasionally causing adjacent structures to merge. To achieve noise reduction while maintaining scale-aware performance, we implemented NAHD filtering within the FWI process, utilizing its hybrid diffusion capabilities. Initially, we employed the CED filter scheme to enhance flow-like structures and remove speckle noise from the FWI gradient, as indicated by the black arrows in Figure 7c. The CED scheme effectively balanced structure preservation with noise reduction, delivering a clearer result compared to the Gaussian-filtered approach. Specifically, CED maintained awareness of small-scale structures and enhanced them while efficiently eliminating incoherent speckle noise from the FWI gradient. Figure 7d presents the results of the SAEP-FWI employing a hierarchical strategy, where the NAHD filter transitioned from CED to EED as the frequency groups progress from low to high. This hierarchical SAEP-FWI strategy exceled in preserving edges, beginning with the application of CED. It enhanced strong coherence in the structural formations while reducing speckle noise. As a result, the main layering information appeared more continuous and clearer than in both Figure 7b,c. From these findings, we conclude that our SAEP-FWI approach significantly improves the quality of the crosshole velocity tomogram in field tests, demonstrating its efficacy in enhancing subsurface imaging in complex environments. This advanced methodological integration within FWI paves the way for more accurate geological assessments and can potentially refine subsurface exploration and monitoring techniques.

To objectively assess the efficacy of the SAEP-FWI algorithm, we analyzed the FWI results alongside logging velocity curves, as depicted in Figure 8. This figure illustrates the reconstructed velocity profiles obtained from the crosshole field dataset using various inversion methods, including the proposed SAEP-FWI method with NAHD filtering. The velocity structures, as inferred from these profiles, were benchmarked against the ground truth, which was represented by both the true model and the initial model. The graphical representation indicates that SAEP-FWI, employing a hierarchical filtering scheme, yields a velocity profile that closely approximates the velocity structures of the true model. This outcome highlights the superior capability of SAEP-FWI in capturing the intricate subsurface velocity variations compared to conventional FWI, both with and without Gaussian filtering, and even FWI utilizing NAHD with CED. The enhanced fidelity of the SAEP-FWI reconstructed velocity profile suggests that the integration of the NAHD filter effectively elevates the quality of the inversion and resolves fine-scale features that are otherwise obscured or misrepresented in the models produced by other methods. Consequently, the SAEP-FWI with NAHD demonstrates a more accurate reconstruction of crosshole velocity structures, affirming its potential as a robust tool in geophysical research for improved subsurface imaging. This analysis substantiates the proficiency of SAEP-FWI in delineating geologically complex layers, ensuring that the velocity models derived from the inversion processes are both reliable and reflective of the true subsurface characteristics. The results also highlight the critical role of advanced filtering techniques in enhancing the resolution and clarity of geophysical imaging, thus supporting more informed decisions in exploration and monitoring activities.

## 4. Conclusions

In this study, we introduce a scale-aware edge-preserving SAEP-FWI algorithm that incorporates an NAHD filter. The NAHD filter is meticulously engineered to mitigate prevalent noise-like anomalies within the conventional FWI gradient computations, while concurrently enhancing structural coherence and sharply defining edges. A key characteristic of the NAHD filter is its dynamic adaptability, which allows it to alternate between coherence-enhancing and edge-enhancing modes based on the local attributes of the images, facilitating precise anisotropic diffusion.

This strategic implementation of the NAHD filter into the FWI framework enables the detailed resolution of minute-scale structures and the refinement of their boundaries, significantly diminishing the noise typically associated with the nonlinear nature of FWI and noisy datasets. The effectiveness of the SAEP-FWI algorithm has been rigorously validated through both experimental setups and real-field applications, demonstrating its ability to significantly improve inversion results. The algorithm excels in suppressing noise and preserving the integrity of structural details and edges, thereby highlighting its potential to substantially enhance the accuracy of subsurface imaging in geophysical research. This contribution is expected to pave the way for more reliable and precise subsurface exploration and analysis.

The rigorous validation process includes comparative studies against traditional FWI methods, quantitatively assessing the improvement in signal-to-noise ratio and the clarity of geological interfaces. The SAEP-FWI’s adaptability in handling diverse geological settings further underscores its robustness and utility in complex environments, where standard inversion techniques often fall short. By providing clearer, more accurate images of subsurface structures, the SAEP-FWI method stands to make significant contributions to the fields of resource exploration, environmental monitoring, and geotechnical engineering. This advancement in FWI technology demonstrates a significant step forward in our ability to interpret and utilize geophysical data for effective subsurface management.

## Figures and Tables

**Figure 1 sensors-24-02881-f001:**
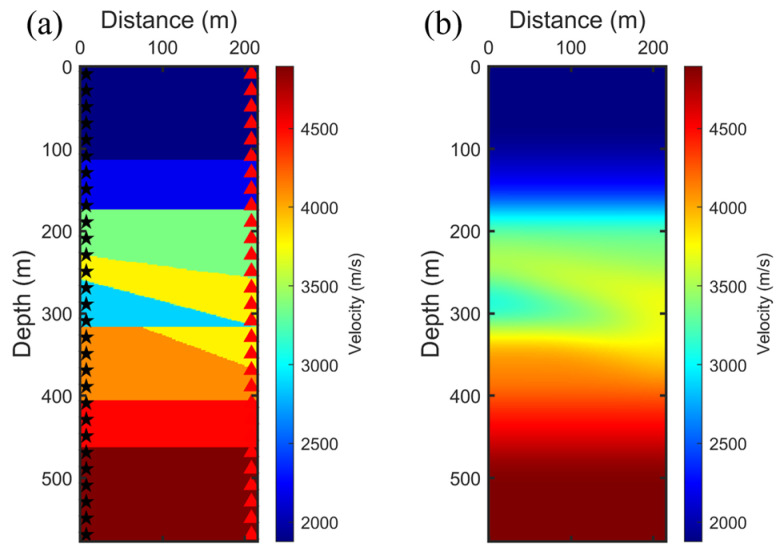
Ground truth of crosshole multi-layer model (**a**) ground truth and (**b**) initial model.

**Figure 2 sensors-24-02881-f002:**
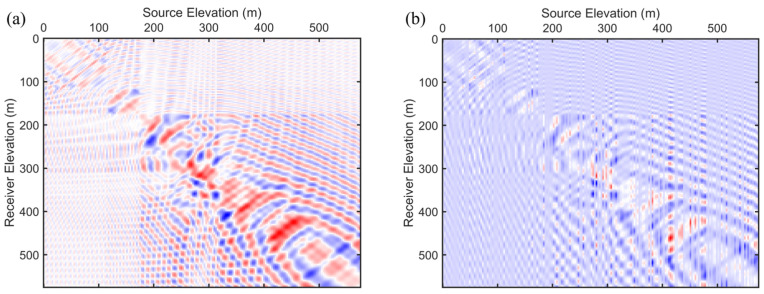
The noisy synthetic dataset (**a**) without noise and (**b**) with noise.

**Figure 3 sensors-24-02881-f003:**
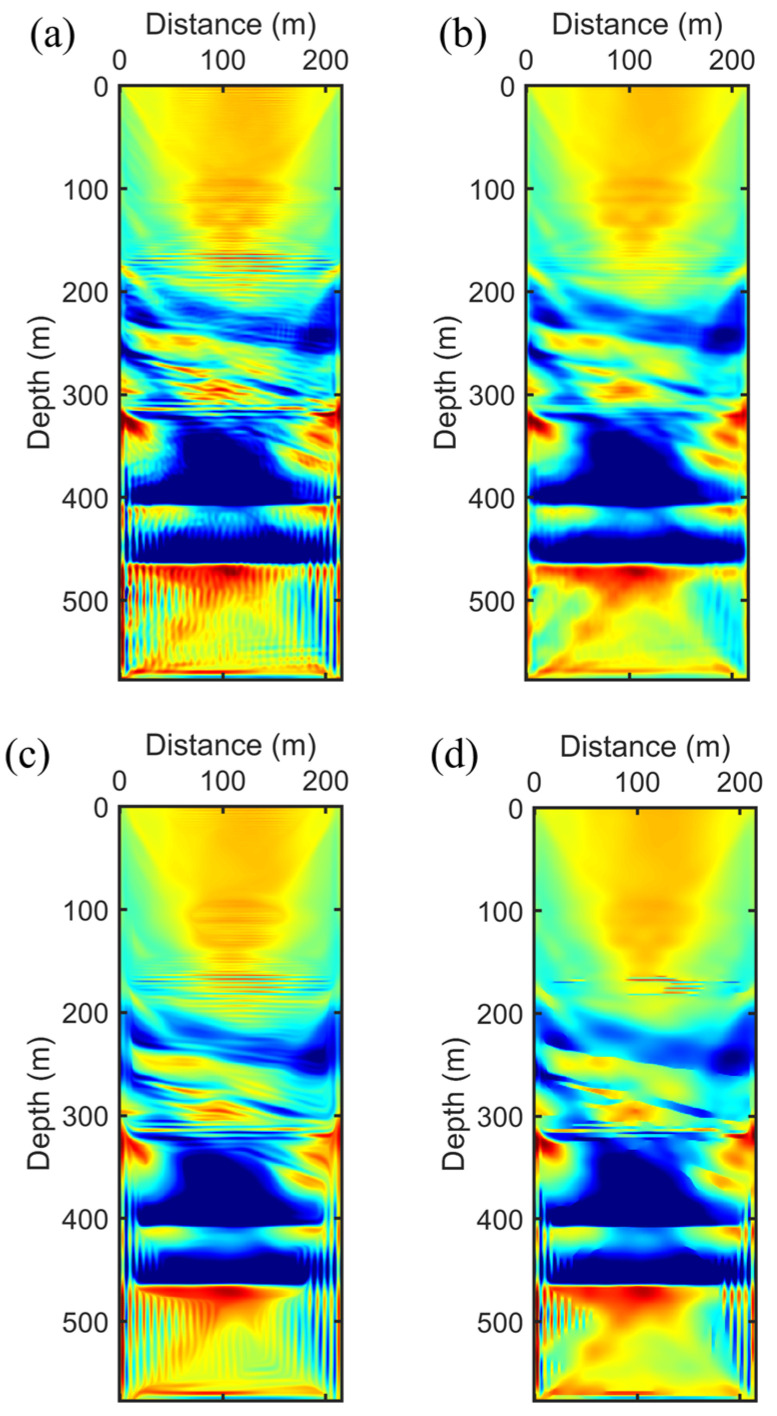
Gradient computed from initial model and its comparison with different filtering: (**a**) Before filtering. (**b**) Gaussian filter. (**c**) NAHD filter switch to CED. (**d**) NAHD filter switch to EED.

**Figure 4 sensors-24-02881-f004:**
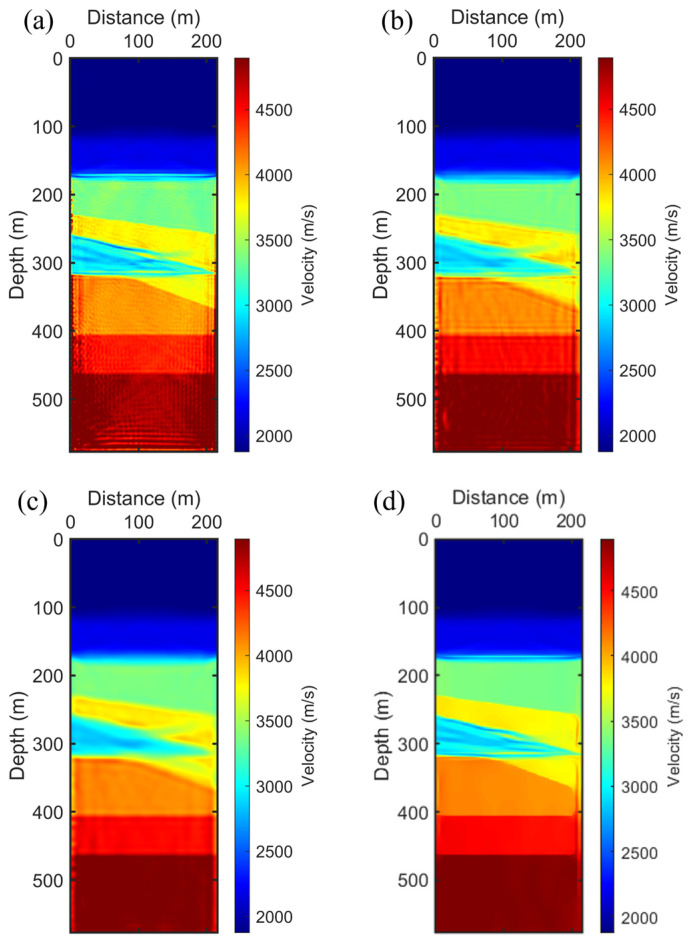
Inversion result: (**a**) FWI without filtering. (**b**) FWI with Gaussian filtering. (**c**) SAEP-FWI with NAHD filtering switch to CED. (**d**) SAEP-FWI with NAHD filtering switch from CED to EED.

**Figure 5 sensors-24-02881-f005:**
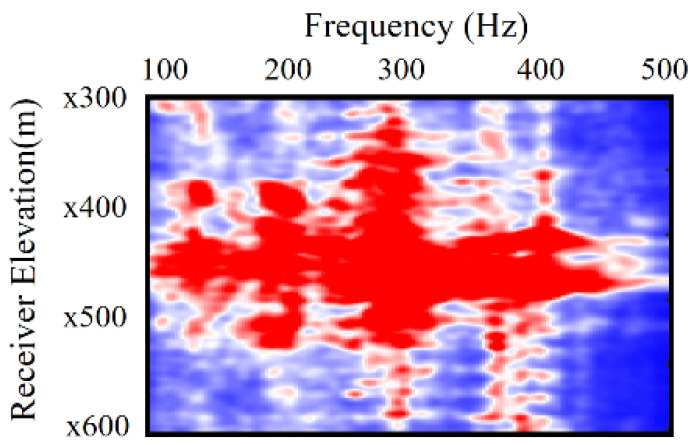
The amplitude spectrum of a typical source gathered at ×450 m.

**Figure 6 sensors-24-02881-f006:**
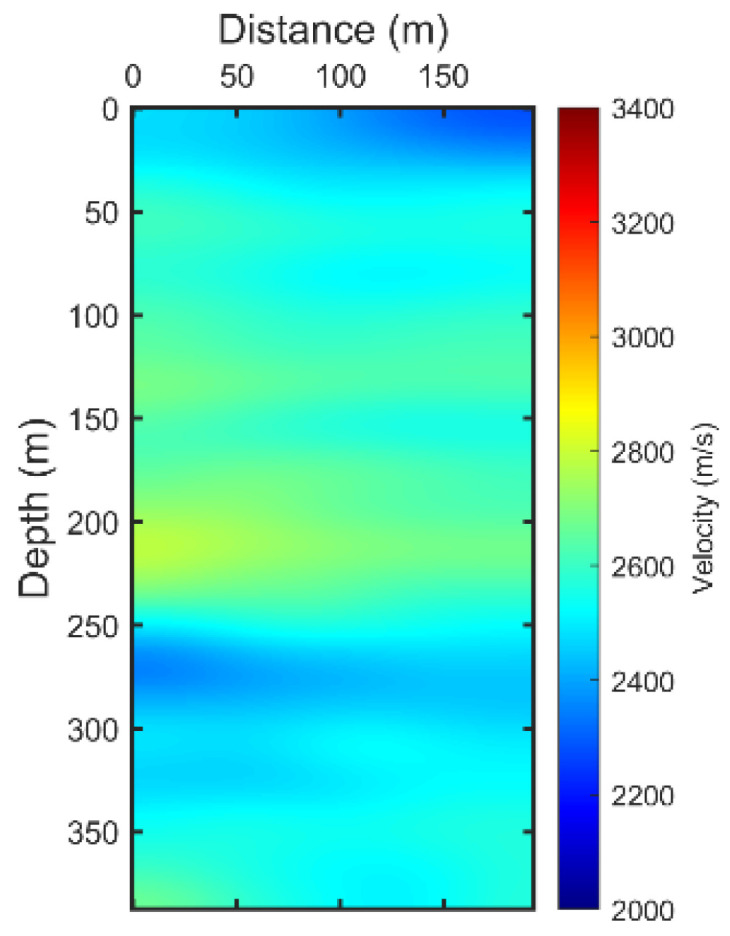
The initial model for FWI.

**Figure 7 sensors-24-02881-f007:**
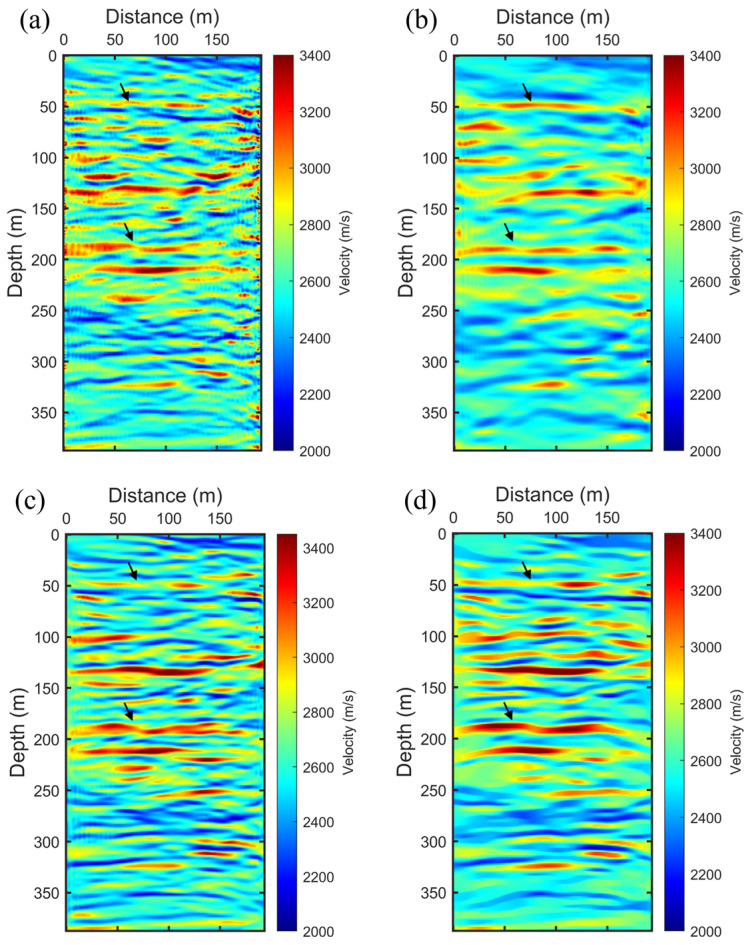
Inversion result: (**a**) FWI without filtering. (**b**) FWI with Gaussian filtering. (**c**) FWI with NAHD filter switch to CED. (**d**) SAEP-FWI with NAHD filtering switch from CED to EED. The image quality improvement is indicated by the black arrows.

**Figure 8 sensors-24-02881-f008:**
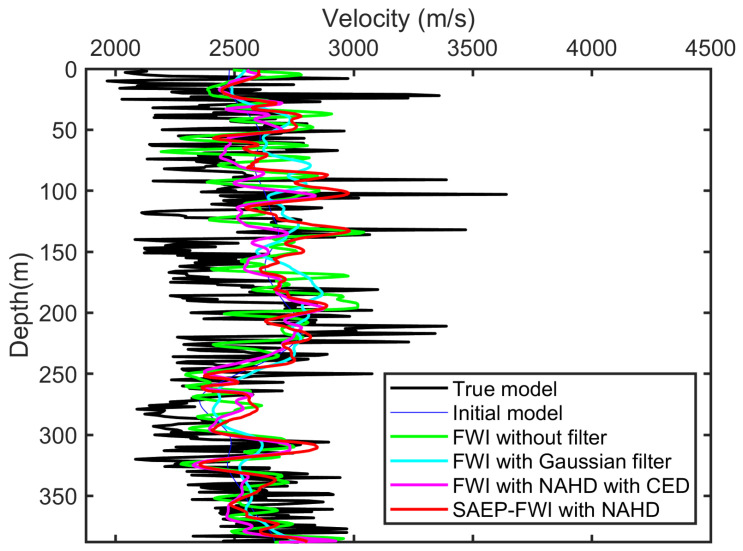
Velocity from logging in the boreholes compared with FWI result.

## Data Availability

Data are contained within the article.

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
