# Peer review of "Scale-Aware Edge-Preserving Full Waveform Inversion with Diffusion Filter for Crosshole Sensor Arrays"

_sensors, 2024, doi:10.3390/s24092881_

Round 1

Reviewer 1 Report

Comments and Suggestions for Authors

This article improves the quality of the inversion image by using a nonlinear anisotropic hybrid diffusion filter to preprocess the gradient and combine it with first-order Tikhonov regularization. My main comments are as follows:

1.    Line 57, P2. What does the ‘offer computational efficiency’ written here mean? Does it mean that the computational efficiency has been improved?

2.    Line 100, P3. The abbreviation (CED)does not correspond to the preceding noun.

3.    Eq. 9 and Eq. 11. There is an error in the gradient expression, please correct it.

4.    Please draw the Misfit function convergence curve

5.    In the numerical example, which frequencies are specifically selected for the iteration frequency group, how many iterations are performed for each frequency group, and whether there is an iteration stop condition.

6.    At which frequency group are the gradients in Fig. 3 calculated?

7.    When the iteration frequency is low, FWI mainly recovers the long-wavelength components of the model. Can you explain how to primarily use CED to preserve small-scale structures while iterating at the initial few lower frequencies?

8.    Line 196-198, P5. The statement of this sentence should be more precise. Although the model becomes more accurate with the iteration process, the increase in iteration frequency makes the full waveform inversion more nonlinear.

Comments on the Quality of English Language

The English expression can be improved, and native speakers are recommended to polish it.

Author Response

Dear Reviewer:

Thank you for the time and effort you put into reviewing the previous version of the manuscript.

We revised the manuscript in accordance with the reviewers’ comments, and carefully proof-read the manuscript to minimize typographical, grammatical, and bibliographical errors. In the document, the modified parts are highlighted.

Reviewer 2 Report

Comments and Suggestions for Authors

The paper discusses an important questoin on how to regularize the gradient within an FWI context. The methods may help in improving FWI implementations, make them more stable ...

The paper might be published if questions, as in the attached file, will be answered sufficient clear.

Good luck

Author Response

(The authors gave the same response as above.)

Reviewer 3 Report

Comments and Suggestions for Authors

The authors of this paper propose a SAEP-FWI algorithm that seamlessly integrates NAHD filters to resolve common noise within traditional FWI gradients and simultaneously enhance structural consistency and sharply defined edges. The authors incorporated NAHD filters into the FWI process to achieve detailed resolution of micro-scale structures and refinement of their boundaries. Both simulation data experiments and field data experiments verified the effectiveness of the SAEP-FWI algorithm. The article has good innovation and practical value, but before the article is published, the following issues need to be considered and modified accordingly.

1 In the simulation experiment, it is recommended to add single-channel velocity profiles with different inversion results to demonstrate the differences in details of the inversion results obtained by different inversion methods.

2  It is difficult to see the superiority of the method proposed in the article from the comparison of the different results and logging in Figure 8. What is the author's explanation for this?

3 How does the filtering method proposed in the article compare with conventional methods in terms of efficiency?

4 Is the filtering method introduced in the article effective in solving the problem of full waveform inversion's dependence on the initial model? It is recommended that the deviation between the initial model setting and the real model be more significant to verify whether the method can help solve the problem of full waveform inversion. Initial model problem.

5 The figures in the article are not marked with numbers a, b, c, d... This is irregular. The author needs to number the subfigures in figures containing multiple subfigures.

Author Response

(The authors gave the same response as above.)

Round 2

Reviewer 1 Report

Comments and Suggestions for Authors

The authors have revised manuscript according to expert opinions, and I recommend publishing it in "Sensors".

Comments on the Quality of English Language

It is best for English writing to be polished by native speakers to increase the readability of the article.

Author Response

Dear reviewer,

Thank you for your review and recommending our manuscript for publication in "Sensors". We appreciate the expert opinions that our revisions and are pleased to hear that the changes meet your expectations.

Thank you once again for your guidance and support.

Best,

Jixin

Reviewer 3 Report

Comments and Suggestions for Authors

The article has been revised by the author and has met the publication requirements. It is recommended to accept it.

Author Response

Dear reviewer,

Thank you for informing us about the acceptance recommendation of our revised manuscript. We are grateful for the opportunity to improve our work based on the feedback received and are delighted to hear that it now meets the publication requirements.

Thank you again for your support and guidance throughout this process.

Best,

Jixin
